# Nanobody-enabled monitoring of kappa opioid receptor states

Tao Che [1✉], Justin English [1], Brian E. Krumm [1], Kuglae Kim [1], Els Pardon [2,3], Reid H.J. Olsen[1], Sheng Wang [1,7], Shicheng Zhang[1], Jeffrey F. Diberto[1], Noah Sciaky[1], F. Ivy Carroll [4], Jan Steyaert [2,3], Daniel Wacker[1,8] & Bryan L. Roth [1,5,6✉]

Recent studies show that GPCRs rapidly interconvert between multiple states although our ability to interrogate, monitor and visualize them is limited by a relative lack of suitable tools. We previously reported two nanobodies (Nb39 and Nb6) that stabilize distinct ligand- and efficacy-delimited conformations of the kappa opioid receptor. Here, we demonstrate via X-ray crystallography a nanobody-targeted allosteric binding site by which Nb6 stabilizes a ligand-dependent inactive state. As Nb39 stabilizes an active-like state, we show how these two state-dependent nanobodies can provide real-time reporting of ligand stabilized states in cells in situ. Significantly, we demonstrate that chimeric GPCRs can be created with engineered nanobody binding sites to report ligand-stabilized states. Our results provide both insights regarding potential mechanisms for allosterically modulating KOR with nanobodies and a tool for reporting the real-time, in situ dynamic range of GPCR activity.

[1] Department of Pharmacology, University of North Carolina at Chapel Hill School of Medicine, Chapel Hill, NC, USA. [2] Structural Biology Brussels, Vrije Universiteit Brussel (VUB), Brussels 1050, Belgium. [3] VIB-VUB Center for Structural Biology, VIB, Brussels 1050, Belgium. [4] Research Triangle Institute, Research Triangle Park, Durham, NC, USA. [5] Division of Chemical Biology and Medicinal Chemistry, Eshelman School of Pharmacy, University of North Carolina at Chapel Hill, Chapel Hill, NC, USA. [6] National Institute of Mental Health Psychoactive Drug Screening Program (NIMH PDSP), School of Medicine, University of North Carolina at Chapel Hill School of Medicine, Chapel Hill, NC, USA. [7] Present address: State Key Laboratory of Molecular Biology, CAS Center for Excellence in Molecular Cell Science, Shanghai Institute of Biochemistry and Cell Biology, Chinese Academy of Sciences, University of Chinese Academy of Sciences, Shanghai, China. [8] Present address: Department of Pharmacological Sciences and Department of Neuroscience, Icahn School of Medicine at Mount Sinai, New York, NY, USA. ✉email: taoche@email.unc.edu; bryan_roth@med.unc.edu

G protein-coupled receptors (GPCRs) represent the single most abundant and targeted class of druggable proteins in the human genome[1,2]. GPCRs sense an immense array of intra- and extracellular stimuli including neurotransmitters, metabolites, odorants, and drugs[2], and allosterically modulate a variety of transducers including as many as 16 Gα proteins[3] and 4 arrestins to control downstream cellular signaling[4]. It is now recognized that different GPCR agonists and antagonists can stabilize distinct receptor conformations with varying potencies and efficacies thereby differentially activating downstream signaling pathways. The process by which ligands differentially modulate cellular signaling is known as functional selectivity[5], and ligands that display this property are referred to as biased. Recent advances in spectroscopic and crystallographic studies have facilitated the observation of these ligand-specific states supporting the existence of an array of GPCR conformational states[6–8]. The molecular mechanisms by which ligands stabilize these GPCR conformational changes are, however, incompletely understood. A key to understanding these transition mechanisms is to obtain structures of each state, not merely generic inactive and active states.

Nanobodies (Nbs)[9] potentially provide a powerful platform to stabilize specific receptor states. As single-domain antibodies developed from camelids, nanobodies display unique properties such as small-size and high-affinity. Nanobodies have been widely used to stabilize the active states of many GPCRs, including the β2- adrenergic (β2AR), M2-muscarinic (M2R), μ-opioid (MOR), k-opioid (KOR), and angiotensin II type 1 (AT1R) receptors[10–15]. Apart from the recognition and stabilization of specific conformations, nanobodies also act as allosteric effectors and can affect both ligand binding affinity and functional activity[8]. For example, stabilization of an active and inactive conformation and dramatic alterations in ligand affinity at the β2AR by allosteric nanobodies were recently reported[8]. These state-specific nanobodies have also been widely applied as conformational biosensors to monitor GPCR dynamics both in vitro and in vivo[16–19]. The available structures of receptor–nanobody complexes reveal that the specific interfaces frequently overlap, at least in part, with G protein interaction surfaces[20].

We previously reported two allosteric nanobodies against KOR—a positive allosteric nanobody (Nb39), and a negative allosteric nanobody (Nb6)[14]. The KOR has emerged as an important therapeutic target because G protein biased KOR agonists are reported to possess analgesic activity without the liabilities of addiction and respiratory depression associated with conventional opioid analgesics which target the μ receptor[21–24]. As well, KOR antagonists and KOR negative allosteric modulators have been proposed both as potential antidepressants and as treatments for addiction[25–27]. To date, no bona fide allosteric binding sites have been identified in KOR, nor are allosteric modulators available for KOR.

To systematically probe the activities of these nanobodies and determine how they reciprocally regulate ligand binding, we investigate the allosteric range of KOR in the absence and presence of Nb6 and Nb39. While Nb39 stabilizes KOR in a fully active state similar to MOR stabilized by Gi heterotrimer[14,28], we show here that Nb6 stabilizes an inactive state distinct from that stabilized by a neutral antagonist. To better understand where Nb6 binds to the receptor and how it allosterically modulates the ligand binding, we solve the complex structure of the KOR bound to the inverse agonist JDTic in the presence of Nb6. Analysis of the KOR–Nb6 complex reveals a unique nanobody binding interface by which Nb6 negatively allosterically modulates KOR. Remarkably, we find that this site can be transferred to several other GPCRs to provide an engineered and functional allosteric site. Finally, given that Nb6 and Nb39 interact with distinct states

of KOR, we report that they can be used as biosensors to detect ligand-stabilized states in real-time with subcellular resolution in living cells.

## Results

**Molecular pharmacology of allosteric nanobodies.** Previously we reported via a Bioluminescence Resonance Energy Transfer (BRET) approach that KOR agonists promote the association of Nb39 and the dissociation of Nb6 from KOR[14]. A simplified model depicting how agonists affect receptor-nanobody interaction is shown in Fig. 1a. The initial net BRET ratio between KOR and Nb6 in the absence of agonist is high (Fig. 1b), implying that Nb6 preferentially binds to the inactive state of KOR. After exposure to high concentrations of the KOR selective agonists Salvinorin A (Sal A)[29] or Dynorphin A (1–17), conformational changes are stabilized that are no longer favorable for Nb6 binding. Consequently, Nb6 apparently dissociates from the receptor. As also shown in Fig. 1b the KOR selective antagonist JDTic[30] modestly potentiates Nb6 binding as might be predicted from the fact that Nb6 already has stabilized the inactive state. In Fig. 1c, the agonist-induced active state of KOR is recognized by Nb39 as evidenced by enhanced nanobody binding as monitored by BRET. We note that SalA and Dynrophin A display different efficacies for Nb39 with SalA being considerably more efficacious than Dynorphin A. Two factors may account for this observed differential efficacy for Nb39 recruitment between Sal A and Dynorphin A: the first is that the receptor state or conformation is ligand-specific[8] (Dynorphin A may induce an active conformation which is not favorable for Nb39 recognition); the second is that cellular accessibility to the two agonists are different. Pertinent to this second possibility we note that Sal A as a diterpene could quickly diffuse across the membrane and activate intracellular receptors (as supported by confocal imaging in this manuscript), while Dynorphin A as a 17 amino acid peptide, would inefficiently penetrate the cell membrane leading to lesser interactions with intracellular receptors.

To further quantify the apparent allosteric effects of Nb6 and Nb39, multiple assays were conducted to examine how these nanobodies differentially regulate KOR ligand binding and functional activity. First, saturation binding was conducted using a KOR selective agonist radioligand $^3$H-U69,593 in the presence of Nb6 or Nb39. Based on the extended ternary complex model of GPCR activation[31], the coupling of G protein or its mimetic (e.g., a nanobody) increases the number of high-affinity agonist binding sites[8,13,14]. Here we demonstrated that Nb6 reduces $^3$H-U69,593 binding by stabilizing an inactive state of KOR, which is unfavorable for agonist binding, while Nb39 increases radioligand binding by stabilizing an active-like state (Fig. 1d). Secondly, a series of KOR ligands with distinct scaffolds were tested in competition binding assays in the presence of each nanobody to investigate the allosteric range of the two state-specific nanobodies. For the competition binding assays, the radioligand $^3$H-U69,593 is no longer a suitable radiotracer for this assay because, as mentioned earlier, its binding kinetics are also affected by the nanobodies. Another available antagonist radioligand $^3$H-JDTic was also tested, and no significant changes were detected for its binding affinity (Ki) or equilibrium constant (Kd) with KOR in the presence of Nb6 or Nb39 (Fig. 1e, Supplementary Fig. 1a and Fig. 2); $^3$H-JDTiC was thus suitable the studies reported here. For the endogenous ligand Dynorphin A (1–17), a 200-fold higher affinity for KOR in the presence of Nb39 compared to the affinity in the presence of Nb6 was observed (Fig. 1f).

Unexpectedly, the binding affinity of a KOR neutral (or silent) antagonist LY2459989[32] was increased ~6-fold in the presence of

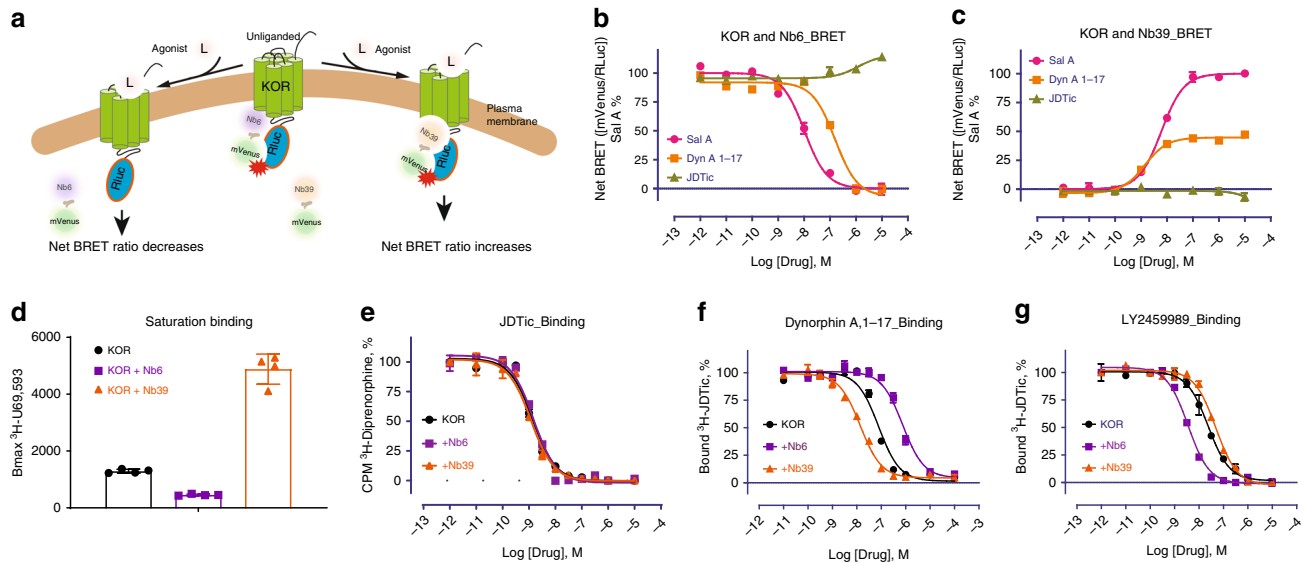

**Fig. 1 Molecular pharmacology of Nb6 and Nb39. a** Schematic of Nb6 and Nb39 interaction with KOR in BRET-based assay. **b** Agonist-induced dissociation of Nb6 from the KOR detected by BRET assay. EC50: Sal A (10.2 ± 0.15 nM); Dynorphin A (155.6 ± 28.6 nM). Note that concentration at −12 represents no-drug added. Net BRET ratio was obtained by the signal$_{(KOR+Nb6)}$ minus signal$_{(KOR)}$. Each data point represents a mean ± s.d. in error bars ($N = 5$, five experiments each done in duplicate) **c** Agonist-induced association of Nb39 with the KOR detected by BRET assay. EC50: Sal A (6.04 ± 0.54 nM); Dynorphin A (1.38 ± 0.17 nM). Each data point represents a mean ± s.d. in error bars ($N = 5$, five experiments each done in duplicate). **d** Saturation binding of the agonist radioligand $^3$H-U69,593 at KOR in the presence of Nb6 or Nb39. Bmax value from the saturation curve was obtained by fitting to the one-site saturation model in Graphpad Prism. c.p.m., counts per minute. Bmax value: 1288 for KOR untreated; 455 for KOR+Nb6; 4880 for KOR+Nb39. ($N = 3$, three experiments each done in duplicate). **e–g** Binding affinity of the antagonist JDTic (**e**), endogenous agonist Dynorphin A (**f**) and neutral antagonist LY2459989 (**g**) in the presence of Nb6 or Nb39. The statistical values are quantified in Supplementary Table 2 by fitting to the "one site-fit Ki" program in GraphPad Prism. Error bars represent SEM. ($N = 3$, three experiments each done in duplicate). Source data are provided as a Source Data file.

Nb6 (Fig. 1g). This phenomenon is reminiscent of a prior observation at the $β_2$AR where Nb60 and Nb80 promoted a full allosteric range for agonists from very-low-affinity to high-affinity states, respectively. Apo (unliganded) receptors exist in multiple inactive states that exchange quickly and thus display conformational heterogeneity[8]. Similarly, for a receptor bound only to an agonist, multiple states such as inactive, intermediate, and active states have been observed, and more complete stabilization of the active state (high-affinity) requires the engagement of transducers (e.g., G proteins or its mimimetic) and agonists[6,7,33]. The increased affinity of LY2459989 supports this observation that the addition of Nb6 to KOR helps stabilize the receptor into a state which is favorable for LY2459989's binding. We also determined the binding of agonist ligands of diverse scaffolds, including the peptide Dynorphin A (1–13), the diterpene Sal A, the epoxy-morphinan MP1104, and the arylacetamide GR89696. As shown in Supplementary Fig. 1b, all displayed the expected effects induced by the positive (Nb39) and negative (Nb6) allosteric nanobodies: their affinities were attenuated by the negative allosteric nanobody Nb6 and enhanced by the positive allosteric nanobody Nb39. One exception was that the agonist MP1104's binding affinity was only slightly affected by Nb6 compared to other agonists, which emphasizes that the allosteric effect is also probe-dependent[34]. MP1104 has displayed very slow off-rate[14] and promotes a specific receptor conformation, which is no longer favorable for Nb6 binding.

Finally, the functional activity of several KOR agonists or partial agonists was evaluated in the presence of Nb6; we previously reported that Nb39 sterically inhibits G protein binding and enhances β-arrestin recruitment[14]. Here we find that Nb6, consistent with its ability to stabilize an inactive state, decreases the potency and/or efficacy of selected KOR agonists (Supplementary Fig. 3). Together, these data reveal that Nb6 and

Nb39 are potent allosteric nanobody modulators that can be used to stabilize inactive and active KOR states. Furthermore, these states can be distinguished from the apo state by a variety of criteria.

**Structure of the KOR–JDTic–Nb6 complex.** As Nb6 promotes a unique receptor state distinct from either the neutral-antagonist-bound or Nb39-stabilized states, we asked if the Nb6-stabilized structure differs from previously reported inactive- and active-like KOR structures[14,35]. To crystallize the KOR–Nb6 complex, the same N-terminal BRIL fusion constructs like the one previously engineered to obtain KOR–Nb39 active-state structure[14] was used to avoid any modification at the intracellular regions of the receptor. Multiple KOR antagonists were then screened in thermostability assays for their ability to stabilize the complex. Of those tested, the antagonist JDTic[36] afforded maximal stabilization of the KOR–Nb6 complex (Supplementary Fig. 4a) as monitored by size-exclusion chromatography. We found JDTic to be a potent inverse agonist at KOR (Supplementary Fig. 4b) via a generic assay which measures the Gi-mediated inhibition of cAMP production. With more specific BRET-based assays, JDTic appears to differentially decrease the coupling between constitutively active KOR and specific Gi/o family members, with greatest efficacy for Gz, $Gα_{oA}$, and $Gα_{oB}$ proteins (Supplementary Fig. 4c). As JDTic promotes a stable KOR–Nb6 complex, the X-ray structure of KOR–JDTic–Nb6 was determined to 3.3 Å resolution in the space group $P2_122_1$ with two monomers per asymmetric unit (Fig. 2a, Supplementary Fig. 4d and Supplementary Table 1). This observed anti-parallel crystal packing should be attributed to the unique crystallization conditions, and is unlikely to represent possible dimerization under physiological conditions. We observed strong densities for KOR, Nb6, and

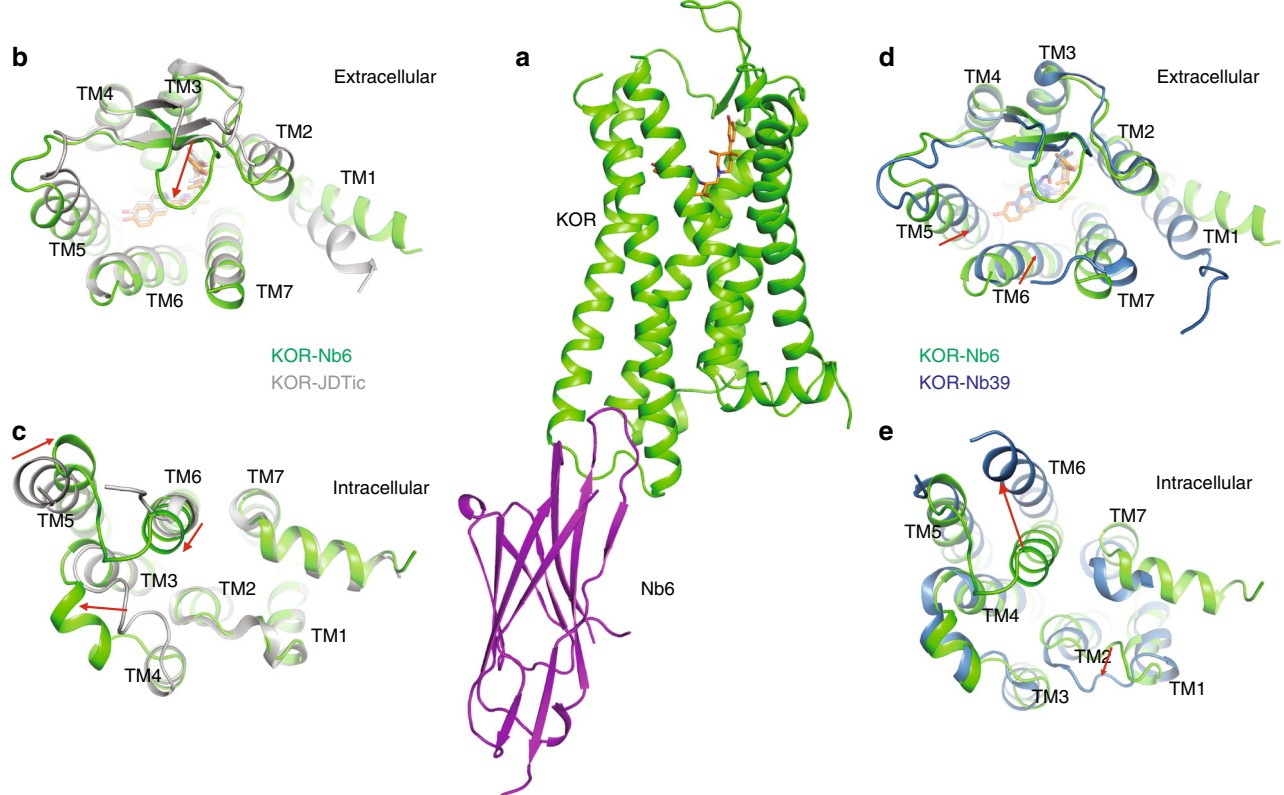

**Fig. 2 Overall alignment of KOR–Nb6 and KOR–JDTic and KOR–Nb39 structures. a** Overall architecture of KOR–JDTic–Nb6 complex. The receptor is shown in green, Nb6 in purple, and JDTic in orange. **b**, **c** Extracellular (**b**) and intracellular (**c**) views of structural alignment of KOR–Nb6 complex (green) and KOR–JDTic inactive state (gray; PDB ID code 4DJH). **d**, **e** Extracellular (**d**) and intracellular (**e**) views of a structural alignment KOR–Nb6 complex (green) and KOR–Nb39 active state (blue; PDB ID code 6B73). Major structural differences are shown as red arrows indicate.

JDTic (Supplementary Fig. 4e). No electron density was observed for the presumably disordered N-terminal BRIL fusion, as was the case for the active state KOR structure[14]. Furthermore, because the two monomers (chain A and B) are nearly identical, we focused on the chain A complex for subsequent analysis.

Together with the previously reported KOR–JDTic (nanobody free)[35] and KOR–Nb39[14] structures, we next analyzed the overall helical movements by comparing them with the KOR–Nb6 structure reported here. The overall structure of KOR–Nb6 is reminiscent of the inactive-state KOR–JDTic structure (Fig. 2b, c) although substantial differences were observed at both extracellular [transmembrane helical (TM) 1 and 5, extracellular loop (ECL) 2 and 3] and intracellular regions [TM5, TM6 and intracellular loop (ICL) 2]. Significantly, ECL2 moves ~6 Å to form an apparent lid on top of JDTic, a feature recently observed in other GPCR structures, including the LSD-bound 5-HT$_{2B}$ serotonin receptor[37] and the risperidone-bound D2 dopamine receptor[38]. The formation of this lid may explain why the binding kinetics (e.g., $k_{on}$ and $k_{off}$) in KOR preincubated with Nb6 were slightly changed ($t_{1/2}$ increases ~2-fold) compared to that in the untreated KOR (Supplementary Fig. 1a). It is also noteworthy that interactions between Nb6 and TM5/TM6 facilitate a mild inward movement of TM6 (~2 Å) by measuring the distance between Cα atoms of D266$^{6.27}$ (superscript denotes Ballesteros–Weinstein numbering). The overall structural similarity between the JDTic-only and Nb6-bound forms is likely because the inverse agonist activity of JDTic stabilizes the receptor in a low-affinity state for agonists, as the binding affinity of JDTic is nearly identical with or without Nb6 (Fig. 1d).

The structural alignment between KOR–Nb6 and active-state KOR–Nb39 revealed canonical active state-like conformational changes and side-chain displacements in the binding pocket and various conserved motifs (Supplementary Fig. 5). Specifically, an inward movement of extracellular TM1, TM5, TM6, and TM7 (Fig. 2d) leads to the contraction of the binding pocket, which is a joint event upon GPCR activation, as also observed in several other Class A GPCR crystal structures[10,11,13,14]. Compared with the active-state KOR, the conformation of ICL3 undergoes significant displacement due to the large outward movement of TM6 (Fig. 2e). Similarly, displacement of intracellular TM5 and TM7 in the active-state KOR is likely a direct effect of TM6[14], and the role of TM6 as a driving force for the expansion of the intracellular part indicates that its conformational differences may play an essential role in determining receptors' coupling specificity for G proteins[39].

**Unique interactions revealed between Nb6 and KOR.** As shown in Fig. 3a, the Complementarity-determining region 3 (CDR3) loop of Nb6 inserts into a crevice between TM5 and TM6, where it forms major interactions with KOR. This interface is distinct from previously reported nanobody-GPCR interfaces (e.g., β$_2$AR-Nb60, β$_2$AR-Nb80, M2R-Nb9-8, MOR-Nb39) and the active-state-like KOR–Nb39 interfaces (Supplementary Fig. 6a), all of which bind via the intracellular core region which is also critical for G protein[11] and/or β-arrestin[40] interactions. From a nanobody perspective, even though the CDR3 adopts a flexible loop conformation, the electron density in this region was sufficient to identify the orientation of side-chains within the residues (Supplementary Fig. 6b). For instance, W106 in Nb6 extends into the crevice and forms hydrophobic interactions with L253$^{5.65}$ and the backbone of R274$^{6.35}$, likely anchoring TM5 and TM6, and

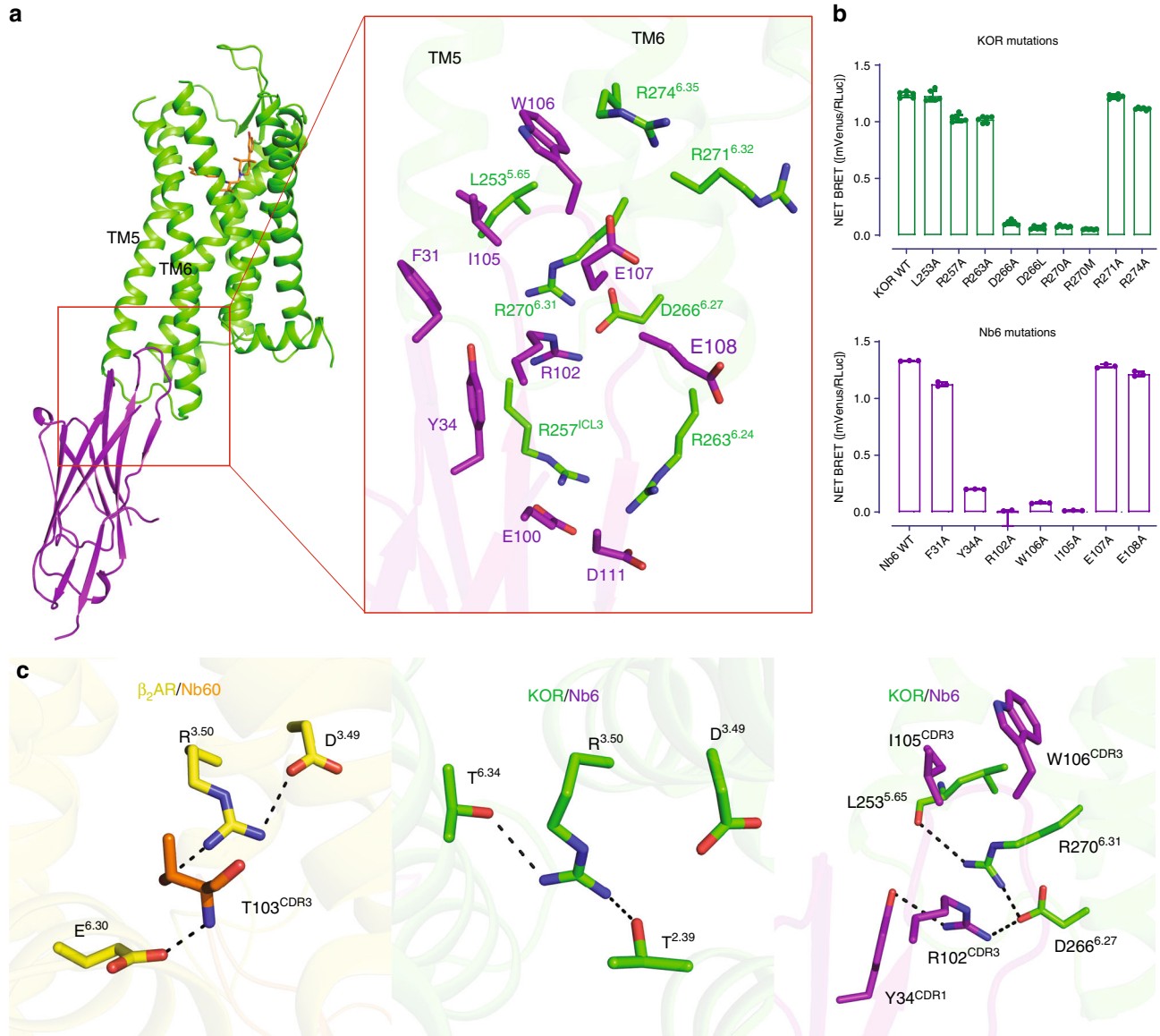

**Fig. 3 Unique interface between KOR and Nb6. a** Details of the interface between KOR (green) and Nb6 (purple) are shown. **b** Mutagenesis studies using BRET identify key residues essential for KOR–Nb6 interactions. Error bars represent SEM. ($N = 3$) **c** left, ionic-lock in β₂AR stabilizing the S2 inactive state; middle, $R^{3.50}$ forms two hydrogen bonds with $T^{2.39}$ and $T^{6.34}$, respectively. These interactions mimic the ionic lock observed in β₂AR; right, R102 in Nb6 mediates a new H-bond network between TM5 and TM6. This interaction is not observed in either KOR–JDTic inactive or KOR–Nb39 active states. Source data are provided as a Source Data file.

preventing the latter from moving outward. R102 in Nb6 forms ionic interactions with two charged residues ($D266^{6.27}$ and $R270^{6.31}$) in the receptor, providing an H-bond network linking TM5 and TM6. All of these interactions appear to contribute to the high affinity of the KOR–Nb6 interaction because mutations of these residues completely abolish the binding of Nb6 to the KOR in BRET assays (Fig. 3b, Supplementary Fig. 6c). Interestingly, Y34 and I105 in Nb6 do not directly interact with the receptor, but mutations of these two residues also attenuate the interaction likely because Y34 and I105 interact with R102 and W106, respectively, assisting the latter two residues to be in the right orientation (Fig. 3b, c). Although mutation of several other residues ($L253^{5.65}$ and $R274^{6.35}$) appears not to affect Nb6 binding to the receptor apparently, they do change the stability of KOR/Nb6 complex as the decreased net BRET ratio and increased potency of agonist Sal A indicate that a weaker interaction between Nb6 and KOR has been formed (Supplementary Fig. 6c).

Although the overall differences between KOR–Nb6 and KOR–JDTic are subtle, there are some changes specific to the Nb6-stabilized state. Apart from the large movement of the ECL2, a twist of the intracellular region of TM5 and TM6 has been observed in the KOR–Nb6 structure (Supplementary Fig. 7a). This twist causes large conformational displacements of side chains of residues in the interface engaged by the CDR3 loop in Nb6 (Supplementary Fig. 7b). The relatively differential position of the TM5 and TM6 also supports that the KOR–Nb6 structure may represent a distinct state from previous KOR–JDTic because the dynamics of TM5 and TM6 has been seen as a major event during receptor transition from inactive to active states[11,28,40]. Because of the stabilization afforded by Nb6, we were able for the first time to visualize an intact ICL3 (i.e., ICL3 is not visualized in previous KOR–JDTic or KOR–Nb39 structures). A comparison of ICL3 between KOR–Nb6 and MOR-Gi structures reveals a significant translocation, probably necessitated accommodating

the Gi protein coupling (Supplementary Fig. 7c). Since Nb6 is a negative allosteric modulator and its binding pocket represents a potentially allosteric site, we continued to investigate how the residues located in the binding pocket contribute to the KOR signaling profile. While alanine mutagenesis shows that most of the residues have minor effects on KOR's G protein activation and β-arrestin translocation, L253$^{5.65}$A and R271$^{6.32}$A appear to significantly reduce receptor activity (Supplementary Fig. 7d). In particular, R271$^{6.32}$A affects the G protein signaling but not arrestin recruitment indicating this residue may play a unique role in KOR-G protein interaction. This is supported by the observation in the closely related MOR-Gi structure that R277$^{6.32}$ in MOR forms an H-bond interaction with the backbone of L353$^{G.H5.25}$ in Gαi1 proteins[28]. The R/K$^{6.32}$ is conserved between different GPCR families: in β$_2$AR, K$^{6.32}$ forms important interactions with Gαs[39] while in class F GPCRs, R$^{6.32}$ acts as a conserved molecular switch to regulate receptor activation and pathway selection[41]. The L253$^{5.65}$A mutation affects both G protein and arrestin signaling simultaneously, indicating that it may play a global role in stabilizing the TM5-TM6 interaction in the active state. Overall, the KOR–Nb6 interface represents an allosteric site distinct from either the G protein or β-arrestin binding site, which may have the potential in the design of KOR allosteric modulators.

Regarding the negative allosteric mechanism of Nb6, we note that in the previously described inactive-state β$_2$AR-Nb60 structure[8], residue T103$^{CDR3}$ in Nb60 acts as a bridge forming an ionic lock between D$^{3.49}$ in TM3 and E$^{6.30}$ in TM6 of β$_2$AR (Fig. 3c, left). This ionic lock has been found conserved in several aminergic GPCRs in their inactive states; when broken this ionic lock appears to be one of the hallmark events of receptor activation[8,42]. However, there is no such ionic bridge observed in opioid receptors as there is a Leu$^{6.30}$ in all opioid receptors, indicating a different mechanism may be involved in the Nb6-mediated KOR inactivation. Here, R$^{3.50}$ in KOR forms two H-bonds with T$^{2.39}$ in TM2 and T$^{6.34}$ in TM6, respectively (Fig. 3c, middle), thereby stabilizing the inactive conformation. Further, R102 in Nb6 forms a hydrogen bond with D266$^{6.27}$, resulting in a new H-bond network formed by D266$^{6.27}$, R270$^{6.31}$, and L253$^{5.65}$ between TM5 and TM6 (Fig. 3c, right). These interactions jointly stabilize an inactive state of KOR by potentiating the interaction between TM5 and TM6 and, importantly, were not observed in the KOR–JDTic or KOR–Nb39 structures. Taken together, these observations suggest that Nb6 stabilizes a different inactive state through combined effects of steric hindrance by W106 and ionic locking by R102 as mutations of these residues abolishes Nb6 binding. Mutations of the critical residues on the interface (R102 or W106 in Nb6) also abolish its inhibitory action for Sal A-mediated cAMP inhibition (Supplementary Fig. 8a).

**Nb6/39 as bidirectional biosensors for opioid receptors**. As shown earlier, Nb6 and Nb39 interact with distinct, non-overlapping receptor conformations. These discrete interaction profiles are potentially ideal for the development of biosensors to probe ligand-stabilized receptor conformations in real-time. To test the applicability of using Nb6 and Nb39 as KOR conformational biosensors, we first used the BRET assay to determine if the Nb6 or Nb39 interactions with KOR were reversible. For these studies, KOR-Rluc (*Renilla* Luciferase) and Nb6-mVenus or Nb39-mVenus were co-expressed in HEK 293T cells. We first tested the ability of Nb6, the inactive-state associated nanobody, to report changes in receptor activity in vitro. The agonist Sal A caused a reasonably rapid ($t_{1/2} = 50 \pm 10$ s) decrease in Nb6-KOR BRET (Fig. 4a), which plateaued after ~10 min. The subsequent addition of the antagonist JDTic rapidly ($t_{1/2} = 75 \pm 12$ s)

increased the BRET ratio equivalent to the level without agonist treatment. JDTic has a much higher binding affinity than Sal A at wild type KOR (Kd, 0.065 nM for JDTic vs 10.9 nM for Sal A) (Fig. 1d, Supplementary Fig. 1b). JDTic also has a slower off-rate in the presence of Nb6. Conversely, Nb6 coupling destabilizes Sal A binding and thus accelerates the dissociation of the agonist Sal A, as supported by the lower Kd measured compared to the KOR/ buffer group (Supplementary Fig. 8b)

We next evaluated the ability of Nb39, the active-state associated nanobody, to report real-time changes in ligand-stabilized states in live cells in vitro. Here the agonist Sal A immediately enhanced the BRET signal between Nb39 and KOR (Fig. 4b). JDTic rapidly ($t_{1/2} = 107 \pm 15$ s) reversed the KOR–Nb39 association, presumably by stabilizing the inactive state (Fig. 4b). The relatively slower $t_{1/2}$ for JDTic in KOR/Nb39 (~107 s) compared to KOR/Nb6 (~75 s) is likely due to the cooperativity from Nb39 because Nb39 has been shown to significantly increase the binding affinity of Sal A (Supplementary Fig. 8b). This reversible interaction between receptor and nanobody could provide a potential conformation-specific biosensor approach for investigating these transitions for a variety of applications.

We next investigated the association of Nb6 and Nb39 with KOR using confocal microscopy to determine if these ligand-stabilized transitions can be visualized in live cells in real-time with subcellular resolution. Here the distribution in single cells was monitored using KOR-mScarlet and Nb6- or Nb39-mVenus. For the first experiment, KOR-mScarlet and Nb6-mVenus are initially co-localized, as predicted from our previous observations that Nb6 binds ligand-free KOR (Fig. 4c). After treatment with Sal A, Nb6 dissociated from the plasma membrane (where it had been co-localized with KOR) and displayed increased cytosolic localization (Fig. 4c, Supplementary Fig. 9a, b). Subsequent treatment with the KOR selective antagonist JDTic reversed this phenomenon.

Next, when KOR-mScarlet and Nb39-mVenus were visualized, the Nb39-mVenus fluorescence was mainly found in the cytoplasm before drug treatment. This indicates that Nb39 recruitment to plasma membrane receptors does not occur in the absence of agonist (Fig. 4d, left panel) and is consistent with our BRET studies which show minimal basal BRET. Upon Sal A stimulation, Nb39 fluorescence rapidly accumulated at the plasma membrane to be co-localized with KOR-mScarlet (Fig. 4d, middle panel). In addition to the Nb39 localization to the plasma membrane, we also observed high co-localization of Nb39 and KOR in discrete subcellular compartments in the cytoplasm (Fig. 4d, Supplementary Fig. 9c, d), reminiscent of the Golgi apparatus[19]. Indeed, in cells expressing the Golgi marker GalT-RFP (Supplementary Fig. 9e, f), following Sal A treatment Nb39-mVenus rapidly localized to this GalT-RFP cellular compartment. The apparent translocation occurs within a few seconds following Sal A addition, consistent with KOR resident under basal conditions in the Golgi apparatus (Supplementary Fig. 9f). Subsequent exposure to the antagonist JDTic partially reversed this, apparently inducing dissociation of Nb39 from KOR by stabilizing the inactive conformation (Fig. 4d, right panel; Supplementary Fig. 9e). These results indicate that these conformation-specific nanobodies can be used to report both cell-surface and intracellular ligand-stabilized receptor conformations in a time-dependent and reversible fashion. A potential advantage of using Nb6 to probe the inactive state is that, as Nb6 pre-binds to the ligand-free receptors, it allows us to establish a baseline signal followed by a ligand response signal necessary for accurately measuring the amplitude of change. Finally, we note that due to the highly conserved sequence and structure of opioid receptors (Supplementary Fig. 10a), Nb6 also binds to μ, δ and

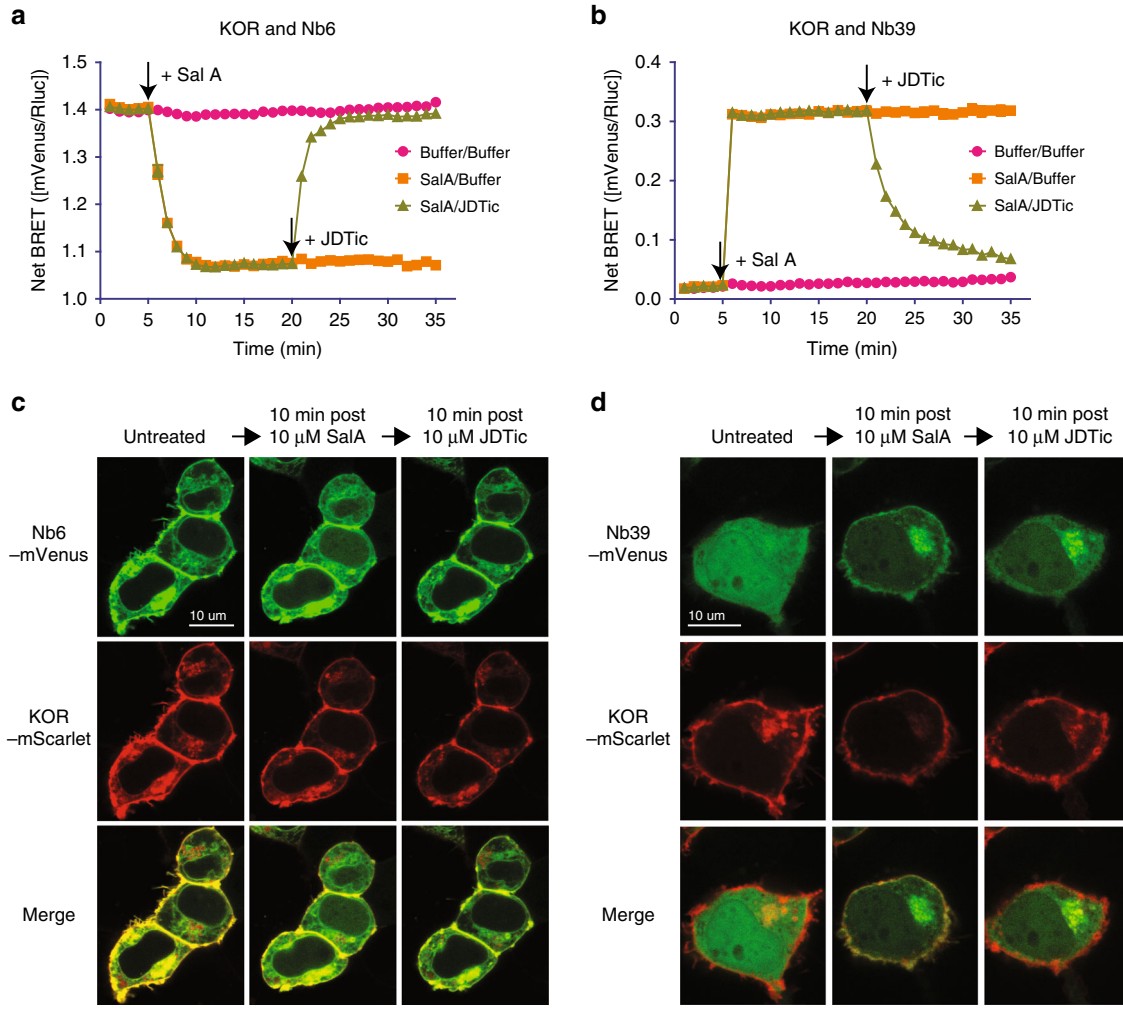

**Fig. 4 Nb6 and Nb39 as biosensors probe KOR conformation in real-time. a, b** Association and dissociation kinetics of KOR–Nb6 (**a**) or KOR–Nb39 (**b**). The experiments were conducted using BRET-based assays and data were analyzed by GraphPad Prism. 10 μM agonist Sal A or antagonist JDTic was added at indicated time points by black arrows. (N = 5, five experiments each done in duplicate) **c** Confocal live-cell imaging of HEK 293T cells expressing KOR-mScarlet and Nb6-mVenus. 10 μM Sal A or JDTic was added to the cell plate at indicated time points. (N = 3, three experiments each done in duplicate) **d** Confocal live-cell imaging of HEK 293T cells expressing KOR-mScarlet and Nb39-mVenus. 10 μM Sal A or JDTic was added at indicated time points. Error bars represent SEM. (N = 3, three experiments each done in duplicate). Source data are provided as a Source Data file.

nociceptin opioid receptors with high affinity (Supplementary Fig. 10b). Nb6 thus can also report the activation and inactivation of the nociceptin receptor (Supplementary Fig. 10c), while Nb39 has been shown unable to bind to this receptor[14]. Thus, together with Nb39, this nanobody pair provides a toolset for studying the dynamics of conformational transitions in situ for other members of the opioid receptor family.

**Nb6 as a potentially useful GPCR conformational sensor.** Inspired by Nb6's capability being as a biosensor for opioid receptors and its unique interactions between KOR and Nb6, we predicted that Nb6 might bind to other GPCRs to report receptor conformational transitions by making chimeric GPCRs. Our hypothesis (Fig. 5a) was that swapping out a portion of ICL3 from an arbitrary GPCR with ICL3 from KOR would provide a site for Nb6 binding. Considering that the interaction between Nb6 and KOR requires an intact ICL3, which has been implicated in playing an essential role in G protein/β-arrestin recognition[11,40], we screened for optimal sites for KOR ICL3 insertion which would have a minimal effect on the activity of a generic Class A GPCR. We subsequently tested seven Class A GPCRs which interact with different G protein subfamilies: Gαs (DRD1, 5HT4R and HRH2),

Gαi (NTS1R and 5HT5AR), and Gαq (5HT2AR and ETA). The full sequences for every chimeric receptor, after optimization, are provided in the Supplementary document. In Fig. 5b, we present data obtained with the Gαq-coupled human 5-HT2A serotonin (5-HT2AR) and ETA endothelin (ETA) receptors, both of which are important drug targets found in a subfamily of GPCRs quite distinct from opioid receptors. Both wild-type 5-HT2AR and ETA, did not display measurable interaction with Nb6 in the BRET assay (Fig. 5b). When their ICL3 regions were replaced by the KOR ICL3 (see details in Supplementary document), Nb6 was now able to bind to 5-HT2AR or ETA as monitored by BRET. Remarkably, their endogenous agonists (5-HT or endothelin-1) caused the dissociation of Nb6 from the chimeric 5-HT2AR or ETA receptors, respectively (Fig. 5b). Importantly, the chimeric receptors remain functional as quantified by G protein activation (Fig. 5c). Other chimeric receptors (D1-dopamine, 5-HT4-serotonin, H2-histamine, NTSR1-neurotensin and 5-HT5A serotonin) were investigated in a similar manner, and they all could respond robustly to respective endogenous ligand without significantly affecting G protein coupling (Supplementary Fig. 11 and 12).

Given that other biosensors such as the miniG proteins have been frequently applied to monitor receptor activation[43], here we

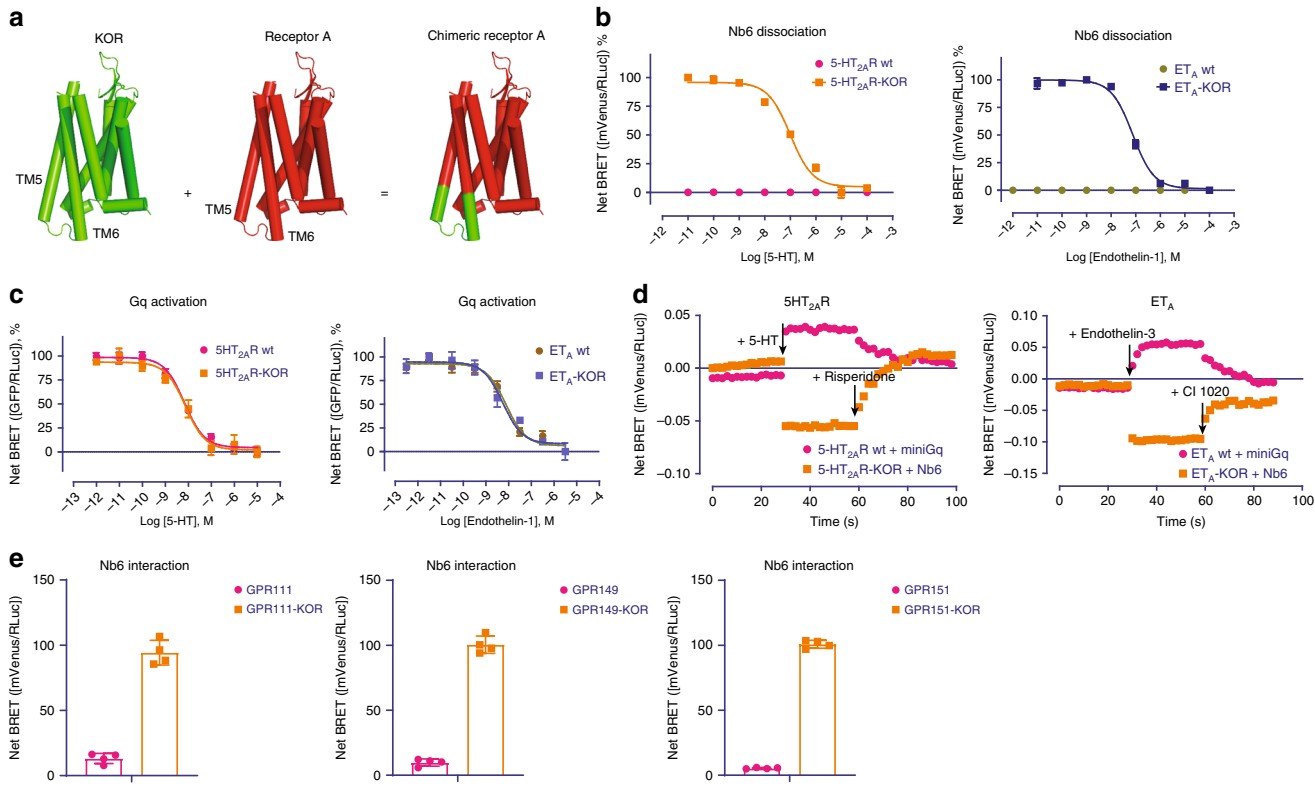

**Fig. 5 The chimeric approach allows Nb6 to bind to 5-HT$_{2A}$R and ET$_A$. a** Schematic of GPCR-KOR chimera generation. **b** BRET assays showing that 5-HT$_{2A}$R-KOR or ET$_A$-KOR chimera can bind to Nb6. Agonist 5-HT or Endothelin-1 causes dissociation of Nb6 from the chimeric receptor 5-HT$_{2A}$R or ET$_A$, respectively. EC50 values were summarized in Supplementary Table 4. ($N = 3$, three experiments each done in duplicate). **c** BRET assays showing that chimeric receptor 5-HT$_{2A}$R-KOR or ET$_A$-KOR can activate canonical G protein coupling similarly compared to the wild type receptors. EC50 values were summarized in Supplementary Table 5. ($N = 3$, three experiments each done in duplicate). **d** A comparison between Nb6 and miniGq proteins in reporting activation and inactivation in 5-HT$_{2A}$R or ET$_A$ in the presence of agonists and antagonists. Experiments were conducted in the BRET assay. The first and second arrow indicates the time when the agonists and antagonists were added, respectively. For different receptors tested, the agonist and antagonist were 5-HT/Risperidone for 5-HT$_{2A}$R, and Endothelin-3/CI 1020 for ET$_A$. The reason that Endothelin-3 was used for ET$_A$ here instead of Endothelin-1 is that Endothelin-1's affinity is too high to be replaced off. **e** BRET assays show that Nb6 can also bind to 'orphan' GPCRs in their basal state. ***$p < 0.001$ as compared to wild type group, unpaired t test. Error bars represent SEM. ($N = 4$, four experiments each done in duplicate). Source data are provided as a Source Data file.

also wondered if Nb6 could report receptor activation in as robust a manner as miniG proteins. Nb6 and miniG were tested in parallel to monitor receptor activation and inactivation in the presence of agonists and antagonists, respectively. As shown in Fig. 5d, Nb6 and miniGq perform comparably at detecting the active and inactive states of 5-HT$_{2A}$R or ET$_A$ receptors. Interestingly, the miniGi occasionally failed to monitor receptor activation (e.g., 5HT$_{5A}$R) by the endogenous ligand 5-HT, and the miniGs proteins were unable to report receptor inactivation by antagonists (Supplementary Fig. 13). Encouraged by this success, we created additional chimeric receptors with several under-studied "orphan" GPCRs (e.g., GPR111, GPR149, and GPR151). Nb6 displayed robust interaction with these chimeric receptors in their basal state, while the wild type receptors do not associate with Nb6 (Fig. 5e). These examples and observations with seven different GPCRs imply that this chimeric approach could be used to faithfully interrogate ligand-stabilized states for the majority of Family A GPCRs for which such nanobodies are unavailable.

## Discussion
Here we show how two KOR state-specific nanobodies can be used as real-time reporters for monitoring both ligand-dependent and ligand-independent conformational states in vitro and in situ. To illuminate the mechanism by which one of these–Nb6–stabilizes an uniqueinactive state, we obtained the crystal structure of KOR

bound to this inactive-state stabilizing nanobody. Together with the previously determined KOR structures with an antagonist JDTic and the Nb39-stabilized active state with MP1104, a comprehensive structural analysis of conformational changes related to KOR activation and inactivation was revealed. Finally, we provide a potential platform that affords real-time in vitro and in situ monitoring of ligand-stabilized GPCR conformations using Nb6 and chimeric GPCRs.

With regard to the conformational transitions of KOR, our data imply that the so-called unliganded or "apo" state as it exists in transfected cells likely consists of a mixture of active and inactive states—each of which can be differentially stabilized by conformation-selective nanobodies. Intriguingly, we find that Nb6 stabilizes a very low-agonist-affinity state reminiscent of that stabilized by the β$_2$AR nanobody Nb60[8]. Significantly, although they both stabilize a very low-affinity inactive state, they do so by interacting with entirely different receptor domains. Thus, for instance, an ionic lock formed in both the inactive β$_2$AR and adenosine A2A receptors was reported to determine the conversion between S1 and S2 inactive states[33,42]. In the β$_2$AR stabilized by Nb60, an S2 inactive state has been observed with an intact ionic lock mediated by Nb60[8]. For opioid receptors, there is no such ionic lock, and as a consequence, Nb6 utilizes an entirely different mode for inactive state stabilization. This unique Nb6-KOR binding surface conceivably could provide a platform for

the discovery of allosteric modulators by, for instance, automated docking approaches[44]. On the other hand, the active-state stabilizing nanobodies Nb39 and Nb80 have overlapping binding sites with G proteins and interact with the core of opioid and β2 adrenergic receptors. These findings imply that active-state stabilizing nanobodies might share a conserved mode of action.

The overall structural differences in KOR are subtle in the absence or presence of Nb6. This is likely due to the potent inverse agonist activity of JDTic, which can stabilize the receptor in a very low-affinity state, as supported by its nearly identical affinity in the presence and absence of Nb6. However, for the neutral antagonist LY2459989, Nb6 stabilizes a state which is more favored by LY2459989 compared with the ligand only state. This finding provides further support for the notion that even for the inactive KOR conformation, at least two states likely exist. How these two low-affinity agonist states might interconvert is currently unclear. Our observation that certain nanobody-stabilized states are favored by specific ligands (e.g., probe dependence) provides a potential opportunity for the identification of new-scaffold agonists and antagonists, as was recently proposed for other receptor–nanobody complexes[45].

Our finding that merely swapping the cognate regions of KOR with several other GPCRs provides an approach for reporting ligand-induced conformations deserves further comment. Producing nanobodies typically requires purified GPCRs and suitable hosts (llamas or sharks) and then several months of additional work to identify and validate their activity[9]. Typically a mixture of nanobodies is recovered, which must be fully characterized as to specifically for the GPCR target and state. Alternatively, a more recent approach uses a yeast-based display of nanobody libraries, although here too, the effort involved is considerable—especially for identifying those which are state-specific[15,46]. In this paper, we show that for at least several unrelated GPCRs, merely swapping a few residues is sufficient to afford robust interactions in a ligand-modulated fashion. Further, given that this simple approach affords the ability to monitor conformational transitions in real-time in vitro and in situ provides the possibility that this could be a relatively general approach. As we have only tested this on Family A GPCRs we do not know if it can ultimately be ported to other GPCR families, although that is currently being investigated.

The technology—although useful—is not without potential confounds. We initially were concerned that this type of swap would alter the signaling ability of the chimeric receptors. In at least the seven cases investigated here, we have not been able to measure any significant degree of attenuation or potentiation of G protein interactions. Certainly, however, future users of the approach will need to verify that signaling, ligand recognition and so on are unaltered before proceeding further with the technology. We also note that the kinetics of ligand-induced alterations in nanobody interactions are rapid and that given the limitations of our current BRET-based readers, we were not able to calculate accurate kinetic constants for some of these interactions.

Finally, we note that KOR has emerged as a primary drug target for medication development for drugs devoid of the side-effects of conventional opioids; such drugs include nalfurafine and buprenorphine. How these drugs activate opioid receptors and produce differential cellular effects are still understudied due to a lack of specific tools. Recently Stoeber et al. used a nanobody-based biosensor with a different nanobody (Nb33) to investigate actions of different opioid drugs in neurons with subcellular resolution[19]. Ligand-specific activation patterns were revealed at a high level of spatiotemporal resolution, specifically between endogenous peptides and non-peptide ligands. The availability of both inactive and active nanobodies for KOR (Nb6 and Nb39, respectively) now enables us to look at a full dynamic range of opioid receptors in a variety of cellular environments.

Additionally, our chimeric receptor approach potentially provides a robust and straightforward platform for investigating these phenomena with a large number of unrelated GPCRs.

## Methods

**Crystallization of KOR–JDTic–Nb6 complex.** Thenanobodies used in this work were generated in Ilama and the detailed method has been illustrated in Che et al.[9,14]. Plasmids Nb6 and Nb39 were transformed to *E. coli* WK6 cells (provided by the Steyaert lab) and were then expressed in the periplasm and purified following steps 70–73[9]. Nanobodies were concentrated and desalted to the buffer: 10 mM HEPES, 100 mM NaCl, and 10% Glycerol and stored at −80 °C for future use.

For crystallography trials, we used a fusion protein of human KOR with an amino-terminal apocytochrome b$_{562}$ RIL (BRIL)[14]. The BRIL-KOR fusion protein was expressed and purified from Sf9 insect cells (Expression Systems, #94-001S). The purified receptor was incubated with 10 µM JDTic (final concentration) all the time during purification. The N-terminal 10× His-tag was removed by the addition of His-tagged TEV protease (Homemade) and incubation overnight at 4 °C. Protease, cleaved His-tag and uncleaved protein were removed by passing the suspension through equilibrated TALON IMAC resin (Clontech) and collected the flow-through. Excessive Nb39 (KOR/Nb39 molar ratio: 1:2) was then added to the protein sample and incubated for 3 h. KOR–JDTic–Nb6 complexes were then concentrated to ~30 mg/ml with a 100 kDa molecular weight cut-off Vivaspin 500 centrifuge concentrator (Sartorius Stedim).

KOR–JDTic–Nb6 complexes were reconstituted into the lipidic cubic phase (LCP) by mixing protein solution and a monoolein/cholesterol (10:1 w/w) mixture in a ratio of 2:3 v/v (protein solution/lipid) using the twin-syringe method[47]. Crystallization was set up in 96-well glass sandwich plates (Marienfeld GmbH) using 50 nl LCP drops dispensed from a 10 µl gas-tight syringe (Hamilton) using a handheld dispenser (Art Robbins Instruments) and overlaid with 1 µl of precipitant solution. Upon optimization, KOR–JDTic–Nb6 crystals were obtained in 100 mM Tris pH7.0, 360–400 mM Ammonium citrate dibasic, 28–32% PEG400. Crystals grew to a maximum size of 50 µm × 40 µm × 30 µm within three days and were harvested directly from the LCP matrix using MiTeGen micromounts before flash-freezing and storage in liquid nitrogen.

**Data collection and structure solution and refinement.** X-ray data were collected at the 23ID-B and 23ID-D beamline (GM/CA CAT) at the Advanced Photon Source, Argonne, IL using a 10 µm minibeam at a wavelength of 1.0330 Å and a Dectris Eiger-16m detector and Dectris Pilatus3-6m detector, respectively. Diffraction data were collected by exposing the crystals for 0.5 s to an unattenuated beam using 0.5° oscillation per frame. The resulting diffraction data from 31 crystals were indexed, integrated, scaled, and merged using HKL2000[48]. Initial phases were obtained by molecular replacement in PHASER[49] using 3 independent models of a truncated 7TM portion of KOR (PDB ID: 4DJH), a nanobody Nb6 from the KOR–MP1104–Nb39 complex (PDB ID: 6B73), and the thermostabilized apocytochrome b$_{562}$RIL protein (PDB ID: 1M6T)[50]. Two copies of the 7TM portion of each the receptor and the nanobody Nb6 but no BRIL were found in the asymmetric unit. Refinement was performed with PHENIX[51] and REFMAC5[52] followed by manual examination and rebuilding of the refined coordinates in the program COOT[53] using 2mF$_o$ - DF$_c$ and mF$_o$ - DF$_c$ maps. The data collection and refinement statistics are shown in Supplementary Table 1. The structural alignment and other analyses were performed in Pymol.

**Bioluminescence Resonance Energy Transfer (BRET) assay.** To measure KOR-nanobody recruitment, HEK 293T cells (ATCC, #CRL-11268) were co-transfected in a 1:3 ratio with human KOR containing C-terminal *Renilla* luciferase (*R*Luc8) and nanobody containing a C-terminal mVenus. After at least 16 h, transfected cells were plated in poly-lysine coated 96-well white clear bottom cell culture plates in plating media (DMEM + 1% dialyzed FBS) at a density of 25–50,000 cells in 200 µl per well and incubated overnight. The next day, media was decanted and cells were washed twice with 60 µL of drug buffer (20 mM HEPES, 1X HBSS, pH 7.4), then 60 µL of the *R*Luc substrate, coelenterazine h (Promega, 5 µM final concentration in drug buffer) was added per well, incubated an additional 5 min to allow for substrate diffusion. Afterward, 30 µL of the drug (3×) in drug buffer (20 mM HEPES, 1X HBSS, 0.1% BSA, pH 7.4) was added per well and incubated for another 5 min. Plates were immediately read for both luminescence at 485 nm and fluorescent mVenus emission at 530 nm for 1 s per well using a Mithras LB940 multimode microplate reader. The ratio of mVenus/*R*Luc was calculated per well and the net BRET ratio was calculated by subtracting the mVenus/*R*Luc per well from the mVenus/*R*Luc ratio in wells without Nb-mVenus present. The net BRET ratio was plotted as a function of drug concentration using the log(agonist) vs. response model in Graphpad Prism 5 (Graphpad Software Inc., San Diego, CA). Equation is shown below:

$$Y = \text{Bottom} + \frac{\text{Top} - \text{Bottom}}{1 + 10^{(\text{LogEC50}-X)}}, \tag{1}$$

where $X$ is the log of dose or concentration; $Y$ is the Response, increasing as $X$ increases; Top and Bottom are the Plateaus in the same units as $Y$, and logEC50 is the same log units as X.

To measure the Gα proteins (Gα$_s$, Gα$_q$, Gα$_{i/o}$, Gα$_z$) mediated activation, HEK 293T cells were co-transfected with the receptor, Gα-Rluc, Gβ$_3$ and Gγ$_9$-GFP in a ratio (5:1:5:5); to measure the mVenus-arrestin-1 or −2 translocation, HEK 293T cells were co-transfected with receptor-Rluc and mVenus-arrestin in a ratio (1:5). After transfection 24 h, cells were plated in the 96-well plates, and the following steps are the same as KOR-nanobody recruitment assay.

To quantify the $t_{1/2}$ in Fig. 4 and Supplementary Fig. 9, data from agonist or antagonist treatment were replotted individually with a non-linear regression model using binding kinetics (Dissociation-One phase exponential decay) equation in Graphpad Prism 5. The equation "Dissociation- One phase exponential decay" is

$$Y = (YO - NS) * e^{-K*X}, \tag{2}$$

where $X$ is time; $Y$ is binding, usually total (constrain NS to 0.0 if specific); Y0 is $Y$ at time 0, in units of $Y$; NS is the non-specific binding, in units of $Y$; $K$ is the rate constant in inverse units of $X$. The half-life is 0.69/K.

**Generation of chimeric receptors**. To make chimeric constructs, different KOR ICL3 DNA sequence was cloned into the receptor-Rluc constructs (human serotonin 5HT$_{2A}$R, 5HT$_4$R, 5HT$_{5A}$R; Dopamine D1 (DRD1); endothelin A (ET$_A$); Neurotensin 1 (NTS$_1$R); Histamine 2 (HRH2)). The detailed DNA and protein sequence information have been provided in the Supplementary document. For different receptors, these insertion sites may vary to obtain a maximum response. These chimeric constructs with C-terminal Rluc tagged were used in BRET assay to test Nb6 recruitment following the BRET protocol; the same chimeric constructs without C-terminal Rluc tagged were used in BRET assays to measure Gα protein activation following the protocols shown above.

To compare the kinetics between chimeric receptor/Nb6 and wild-type receptor/miniG proteins, HEK 293T cells were co-transfected with chimeric receptors with C-terminal Rluc and Nb6-mVenus (GPCR-KOR-Rluc + Nb6-mVenus) or wild-type receptor and mVenus-miniG (GPCR-Rluc + mVenus-miniG) at ratio 1:5, respectively. The transfected cells were treated following the BRET protocol. During the test, luminescence was recorded in the order: background for 30 s, + 10 μM agonist for 30 s, + 10 μM antagonist for 30 s. Cells transfected with GPCR-KOR-Rluc or GPCR-Rluc only are the background and the net BRET ratio will be the recorded value in the experimental groups subtracted by the background. Data were then plotted in Prism 5.

**Radioligand binding assays**. Binding assays were performed using Sf9 membrane fractions expressing the crystallization construct BRIL-KOR or HEK293 T membrane preparations transiently expressing KOR wt or KOR mutants. Binding assays were set up in 96-well plates in the standard binding buffer (50 mM Tris, 0.1 mM EDTA, 10 mM MgCl$_2$, 0.1% BSA, pH 7.40). Saturation binding assays with 0.1–20 nM [$^3$H]-U69,593 in standard binding buffer were performed to determine equilibrium dissociation constant (Kd) and Bmax, whereas 10 μM final concentration of JDTic was used to define nonspecific binding. For the competition binding, 50 μL each of $^3$H-JDTic (final 0.8 nM), drug solution (3×) and homogenous membrane solution was incubated in 96-well plate in the standard binding buffer in the absence or presence of purified nanobodies (final concentration 5 μM). Reactions (either saturation or competition binding) were incubated for 2 h at room temperature in the dark and terminated by rapid vacuum filtration onto chilled 0.3% PEI-soaked GF/A filters followed by three quick washes with cold washing buffer (50 mM Tris HCl, pH 7.40) and read. Results (with or without normalization) were analyzed using GraphPad Prism 5.0 using one site—Fit Ki model (Eqs. (3) and (4)):

$$logEC50 = log10^{logKi\left(1 + \frac{[Hot]}{Kd}\right)} \tag{3}$$

$$Y = Bottom + \frac{Top - Bottom}{1 + 10^{(X - LogEC50)}} \tag{4}$$

where $X$ is the log molar concentration of unlabeled ligand; $Y$ is the binding in any convenient units.

Top and Bottom are the plateaus in units of $Y$-axis; log$K_i$ is the log of the equilibrium dissociation constant of tested ligand; [Hot] is the concentration of radioligand, and Kd is the equilibrium dissociation constant of radioligand.

**Ligand association and dissociation assays**. Binding assays were performed using membrane preparations of HEK-293T cells transiently expressing human wild-type KOR at room temperature. The same aliquots of membranes were preincubated with buffer, 5 μM Nb6 or Nb39 proteins for 30 min. Radioligand dissociation and association assays were performed in parallel using the same concentrations of radioligand, membrane preparations and binding buffer (50 mM Tris, 10 mM MgCl2, 0.1 mM EDTA, 0.1% BSA, pH 7.4). All assays used at least two concentrations of radioligand (0.6–1.5 nM[$^3$H]-JDTic. For dissociation assays, membranes were incubated with radioligand for at least 2 h at room temperature before the addition of 10 μl of 10 μM excess cold ligand JDTic to the 200 μl membrane suspension at designated time points. For association experiments, 100 μl of radioligand was added to 100 μl membrane suspensions at designated time points.

Time points spanned 1 min to 6 h, depending on experimental conditions and radioligand. The association (Kon) and dissociation (Koff) values were calculated using the Prism model:

Association kinetics to obtain kon

$$Y = Ymax * (1 - e^{-kob*X}). \tag{5}$$

Dissociation—one phase exponential decay to obtain koff

$$Y = (YO - NS) * e^{-K*X}, \tag{6}$$

$X$ is time; $Y$ is binding, usually total (constrain NS to 0.0 if specific); Y0 is $Y$ at time 0, in units of $Y$; Ymax is the maximum binding in $Y$ units; NS is the binding at very long times, in units of $Y$; $K$ is the rate constant in inverse units of X. The half-life is 0.69/K.

To use an alternative assay to measure JDTic kinetics in the presence of Nb6 or Nb39 as shown in Supplementary Fig. 2. The kon was determined by incubation of KOR cell membranes with the indicated concentrations of [$^3$H]-JDTic (200, 400, 800 pM) for various time periods. Assays were set up the same as the above association assay. Data were fitted using a one-phase exponential association function to yield an observed on-rate (k$_{ob}$) (Eq. 4). kob was plotted against the [$^3$H]-JDTic concentration, employed for indirect determination of koff (y-intercept = 0) (Eqs. (7) and (8)):

$$kob = kon * [Hot] + koff, \tag{7}$$

$$kd = \frac{koff}{kon}. \tag{8}$$

For the determination of kon and koff for unlabeled Sal A, membranes containing either wild-type KOR were incubated with [$^3$H]-JDTic and several concentrations of Sal A (1, 10, 100, 1000, 10000 nM). Non specific binding was determined by addition of 10 μM JDTic. Immediately (at time = 0 min), plates were harvested by vacuum filtration onto 0.3% polyethyleneimine pre-soaked 96-well filter mats (Perkin Elmer) using a 96-well Filtermate harvester, followed by three washes with cold wash buffer (50 mM Tris pH 7.4). Scintillation cocktail (Meltilex, Perkin Elmer) was melted onto dried filters and radioactivity was counted using a Wallac Trilux MicroBeta counter (PerkinElmer). Data were analyzed using 'dissociation-one phase exponential decay' or 'association kinetics-two or more concentrations of hot radioligand' in Graphpad Prism 5.0. The previously determined [$^3$H]-JDTic kon and koff rates of KOR (+ buffer or Nb6 or Nb39) were used to estimate the kon and koff rates of Sal A using the 'kinetics of competitive binding' equation in Graphpad Prism 5.0

Kinetics of competitive binding model:

$$Y = Q * \left( K4 * \frac{DIFF}{KF * KS} + \left(\frac{K4 - KF}{KF}\right) * e^{(-kF*X)} - \left(\frac{K4 - KS}{KS}\right) * e^{(-KS*X)} \right). \tag{9}$$

Specifically:

$$KA = K1 * [L] * (1e - 9) + K2$$
$$KB = K3 * [L] * (1e - 9) + K4$$
$$S = \sqrt{((KA - KB)^2 + 4 * K1 * K3 * [L] * I * (1e - 18))}$$
$$KF = 0.5 * (KA + KB + S)$$
$$KS = 0.5 * (KA + KB - S)$$
$$DIFF = KF - KS$$
$$Q = Bmax * K1 * L * \frac{1e - 9}{DIFF}$$

$X$ is time in seconds or minutes; $Y$ is the specific binding in cpm or some other unit.

$K1$ is the association rate of radioligand in inverse molar times inverse time); $K2$ is the dissociation constant of radioligand in inverse time), and $L$ is the radioligand in nM. Bmax is the maximum binding, in units of $Y$-axis is the cold drug in nM.

$K3$ is the on rate of tested drug (inverse Molar times inverse time); $K4$ is the off rate of tested drug (inverse time);

**Confocal Imaging**. To prepare confocal imaging, HEK 293T cells were first plated in MatTek dishes pre-coated with poly-L-Lysine at density 75,000 cells/plate in 10% FBS DMEM. The next day KOR-mScarlet and Nb6-mVenus or Nb39-mVenus were co-transfected with TransIT-2020 transfection reagent. These reporters were co-expressed in HEK 293T cells with a 4:1 ratio to ensure an excess of receptor and minimize free nanobody. For Golgi labeling, KOR, Nb39-mVenus and GalT-RFP were co-transfected. After at least 16 h, the medium was changed to 1% dFBS DMEM and incubated at 37 for another 24 h. On the experiment day, cell medium was removed and changed to drug buffer (20 mM HEPES, 1X HBSS, pH 7.4 supplemented with 0.1% BSA, 0.01% L-ascorbic acid and 4.5 g/L Glucose). Measurements were carried out on an inverted Olympus FV3000RS confocal microscope equipped with a Tokai Hit WSKMX live-cell chamber (set to maintain sample heat at room temperature) and a LUMPL FL 60× SuperCorr Plan ApoN 1.40 NA oil immersion objective and four cooled GaAsP photomultiplier detectors. One-way/ Round trip galvanometer/resonant scanning was used for acquisition with a 1× optical zoom. The 488 nm diode laser was used for the mVenus excitation and the

561 nm diode laser was used to excite the mScarlet or RFP, with the laser power set between 0.1 and 5% according to the power slider in the FV31S-SW microscope software, for monitoring fluorescence intensities. The variable slit had a nominal bandwidth of 500–545 nm for the green emission channel and 590–650 nm for the red emission channel.

All image processing was done using open-source Fiji[54]. To quantify KOR–Nb6 translocation, Binary Cell masks (having values of 255 or 0) were created by thresholding Nb6-mVenus intensity in single-cell images at each time point, using Triangle segmentation and filling all holes were using binary masking function. A set of Cell ring masks were created by successive erosion of mask images by 15 pixels and subtracting inner region masks. One-bit masks (having values of 1 or 0) were multiplied with the original cell images to create 16-bit images for the outer and inner rings. The Ratio of the inner Median pixel (cytosol) value to the outer ring (membrane) pixel value was measured and plotted as a function of time.

To quantify KOR–Nb39 translocation, the Co-localization of Golgi was measured by masking images for bright signals in green (Nb39-mVenus) and red (GalT-RFP) channels. Golgi masks were created by creating masks representing the overlap of these red and green binary masks using the Boolean AND function. The Golgi masks were multiplied with the original NB39-mVenus cell images for each time point, measured and normalized to the peak values.

**cAMP inhibition assay**. To measure KOR/$G_{\alpha i}$-mediated cAMP inhibition, HEK 293T (ATCC CRL-11268) cells were co-transfected with human KOR along with a luciferase-based cAMP biosensor (GloSensor; Promega) and with or without varying concentration of Nb6 DNA, and assays were performed similarly to previously described[55]. After 16 h, transfected cells were plated into Poly-lysine coated 384-well white clear bottom cell culture plates with DMEM + 1% dialyzed FBS at a density of 15,000–20,000 cells per 40 µl per well and incubated at 37 °C with 5% $CO_2$ overnight. The next day, drug solutions were prepared in fresh drug buffer [20 mM HEPES, 1X HBSS, 0.3% bovine serum album (BSA), pH 7.4] at 3X drug concentration. Plates were decanted and received 20 µl per well of drug buffer (20 mM HEPES, 1X HBSS) followed by the addition of 10 µl of drug solution (3 wells per condition) for 15 min in the dark at room temperature. To stimulate endogenous cAMP via β adrenergic-Gs activation, 10 µl luciferin (4 mM final concentration) supplemented with iso-proterenol (400 nM final concentration) were added per well. Cells were again incubated in the dark at room temperature for 15 min, and luminescence intensity was quantified using a Wallac TriLux microbeta (Perkin Elmer) luminescence counter. Results (relative luminescence units) were plotted as a function of drug concentration, normalized to % SalA stimulation, and analyzed using log(agonist) vs. response in GraphPad Prism 5.0 (Eq. 1).

**Tango arrestin recruitment assay**. The KOR Tango constructs were designed and assays were performed as previously described[56,57]. HTLA cells expressing TEV fused-β-Arrestin2 (kindly provided by Dr. Richard Axel, Columbia Univ.) were transfected with the KOR Tango construct with or without varying concentration of Nb6 DNA. The next day, cells were plated in DMEM supplemented with 1% dialyzed FBS in poly-L-lysine coated 384-well white clear bottom cell culture plates at a density of 10,000–15,000 cells/well in a total of 40 µl. The cells were incubated for at least 6 h before receiving drug stimulation. Drug solutions were prepared in drug buffer (20 mM HEPES, 1X HBSS, 0.3% BSA, pH 7.4) at 3X and added to cells (20 µl per well) for overnight incubation. Drug solutions used for the Tango assay were exactly the same as used for the cAMP assay. The next day, media and drug solutions were removed and 20 µl per well of BrightGlo reagent (purchased from Promega, after 1:20 dilution) was added. The plate was incubated for 20 min at room temperature in the dark before being counted using a luminescence counter. Results (relative luminescence units) were plotted as a function of drug concentration, normalized to % SalA stimulation, and analyzed using log(agonist) vs. response in GraphPad Prism 5.0 (Eq. 1).

**Reporting summary**. Further information on research design is available in the Nature Research Reporting Summary linked to this article.

## Data availability

Data supporting the findings of this manuscript are available from the corresponding authors upon reasonable request. A reporting summary for this Article is available as a Supplementary Information file. The accession number for the coordinates and structures factors of KOR–JDTic–Nb6 complex reported in this paper is PDB: 6VI4.

The Source Data underlying Figs. 1b–g, 3b, 4a, and 5b–e and Supplementary Figs. 1a, b, 2, 3a, 4b, c, 6c, 7d, 8a, b, 10b, c, 11–13 are provided as a Source Data file.

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

## Acknowledgements

This work was supported by NIH grants (PO1DA0357634; R37DA045657, and RO1MH112205), the NIMH Psychoactive Drug Screening Program Contract, and the Michael Hooker Distinguished Chair of Pharmacology (to B.L.R.). We thank INSTRUCT, part of the European Strategy Forum on Research Infrastructures (ESFRI), and the Research Foundation—Flanders (FWO) for their support to the nanobody discovery and thank Nele Buys and Katleen Willibal for the technical assistance. R.H.J. Olsen was also supported by grant F31-NS093917. We thank Michael S. Placzek for providing LY2459989. We also thank Roshanak Irannejad for providing the GalT-RFP construct. We gratefully acknowledge M.J. Miley and the UNC Macromolecular Crystallization Core for advice and use of their equipment for crystal harvesting and transport, which is supported by the National Cancer Institute (award number P30CA016086). We thank J. Smith and R. Fischetti and the staff of GM/CA@APS, which has been funded with Federal funds from the National Cancer Institute (ACB-12002) and the National Institute of General Medical Sciences (AGM-12006). This research used resources of the Advanced Photon Source, a U.S. Department of Energy (DOE) Office of Science User Facility, operated for the DOE Office of Science by Argonne National Laboratory (contract no. DE-AC02- 06CH11357).

## Author contributions

T.C. designed experiments, expressed and screened nanobodies, receptor constructs, and receptor ligands, crystallized nanobody–receptor complex, collected diffraction data, refined the structure, performed ligand binding and functional assays, analyzed all data, and prepared the manuscript. J.G.E. helped with confocal live-cell imaging and analyzed the data and prepared the manuscript. B.E.K. designed experiments and helped with diffraction data collection and processed diffraction data. K.K. helped with structure refinement. R.H.J.O. helped with transducerome screening. S.W. helped with crystal optimization. N.S. helped with live-cell imaging analysis and quantifications. J.S. and E.P. immunized llamas, generated nanobody library and selected and identified all nanobodies. S.C.Z. helped characterize the chimeric receptors. I.F.C. provided the JDTic compound. D.W. supervised the structure determination strategy, helped with diffraction data collection, processed the diffraction data, solved and refined the structure. B.L.R. designed the experiments, was responsible for the overall project strategy and management and prepared the manuscript.

## Competing interests

The authors declare no competing interests.
