## [Peer Review File · Nature Communications]

Reviewers' Comments:

Reviewer #1:

Remarks to the Author:

The authors have very extensively revised the manuscript to address my concerns. I support publication of this manuscript in Nature Communications.

Reviewer #2:

Remarks to the Author:

General comments.

Overall, this is a very impressive work. Getting a crystal structure of any GPCR is difficult, especially in a complex with a Nb. These new tools would be undoubtedly extremely useful for research on KOR and related receptors. The manuscript is well written and easy to follow.

I have reviewed this work before, and the authors have addressed most of my comments in the revised version.

My general comment is that the newly revised parts would benefit from a proofreading.

My second comment is that some experiments are still not described in sufficient details. This assumes a very high level of expertise from a person who may want to repeat these experiments and telepathic abilities to connect to the author's brain.

The gold standard that would really increase the reproducibility of the work would be to write everything in sufficient details so that the first year PhD student could repeat them. Same goes for the details of the data analysis. Simply referring to the software build-in functions may not be sufficient. Not everybody has access to this specific software, nor the software used may have all functions defined.

Addressing the following specific comments could further improve the manuscript:

p7. Another available antagonist radioligand 3H-JDTic was also tested, and no significant changes have been found in its binding affinity (K_i) or equilibrium constant (K_d) with KOR in the presence of Nb6 or Nb39 (Fig. 1e, Supplemental Fig. 1a), which makes it an ideal choice to test other KOR ligands.

Fig S1a Shows Kinetic measurements that need to be repeated at several concentrations of the ligand. A standard plot k_{obs} vs ligand concentration should be shown. k_{off} should also be calculated, as well as measured in a separate competition experiment. This is not dissimilar to the data presented on Fig S7 (although there the results of the competition experiment is shown). A good example could be found here: DOI: 10.1007/978-1-4939-8630-9_10.

The details of the data analysis are missing. A reference to a model in Prizm assumes the reader to have access to the same version of Prizm and that their model is unmodified. This is very detrimental to the reproducibility of the results.

Reported K_d is rather low to be measured by 3H ligands.

Is this a kinetic Kd?

The reported values differ by a factor of 10.

it is also referred in text as showing no significant difference in K_i or K_d but it seems to be wrongly referred to as it does not show these data. Did the authors mean S1b?

p8. N6 co should be Nb6

p10 (Supplemental Fig. 1a). supplementary?

p13. backbone of L353 in helix $\alpha 5$ in Gai proteins²⁸. Ideally, CGN numbering (eg, G.H5.25; doi 10.1038/nature14663) should be used for G proteins, or at least the isoform should be indicated as it can be L354 in Gai2 or Gaz, for example.

p15. JD_{Tic} also has a slower off-rate in the presence of Nb6 and appears to outcompete the binding of Sal A, occupies the binding pocket and stabilizes the KOR/Nb6 complex.

p15. The relatively slower $t_{1/2}$ for JD_{Tic} in KOR/Nb39 (~107 s) compared to KOR/Nb6 (~75 s) is likely due to the cooperativity from Nb39 because Nb39 has been shown to significantly increase the binding affinity of Sal A (Supplemental Fig. 1b and 7b)

S1b does not show any kinetic data. Wrong reference to the figure?

p15. "relatively fast kinetics". Relative to what? It is not clear what the authors use as a reference point, or if there are any generally accepted values for kinetics that would qualify nanobody's as conformation-specific tools.

Fig S2a-all panels. Y axis. Looks like the axis title is wrong

Fig S3a Y axis. Wavelength used not shown. A280?

Fig S3b Y axis. Looks like the axis title is wrong

Figure S2. This is unaddressed original comment (ug of what? It has to be clear from the figure or the legend without going through the methods – and this is not explained in the methods well) that the authors do not explain the details of the experiments in sufficient detail. Although they have addressed this in the rebuttal letter they have not incorporated the changes in the manuscript.

Dmitry Veprintsev

Reviewer #3:

Remarks to the Author:

The authors previously reported two nanobodies interacting with the kappa opioid receptor (KOR): Nb6 that decreases agonist binding (but interacting with any opioid receptor subtype), and Nb39 that in contrast increases agonist binding and appears specific for KOR. The present paper was aimed at better characterizing these two nanobodies and identify the structural bases for their allosteric effects. Nb6 is shown to bind preferentially an inactive form of KOR, very similar to that stabilized by the

antagonist JD1ic, then decreasing agonist potency at displacing 3H-JD1ic binding. In contrast, Nb39 increases antagonist and decreases agonist affinities. The effect of Nb6 was confirmed by functional studies showing a decreased potency and/or efficacy of several KOR agonists. The crystal structure of KOR-JD1ic-Nb6 complex was solved, and revealed an inactive conformation, slightly more compact than that observed with a neutral antagonist, and in which the ECL2 form a lid at the binding site entrance. This structure is then quite different from that reported by the authors for an active form of the receptor stabilized by Nb39. These nanobodies are then used to detect KOR states in living cells, with Nb39 interaction being favored upon agonist action, while Nb6 interaction is favored in the basal or antagonist bound forms. They eventually extend their use of Nb6 to detect the activation of other class A receptors based on a chimeric receptor assay.

Minor points

- The authors should add a short comment on the presence of two KORs per asymmetric crystal units, indicating the proteins are upside down, then unlikely to represent of possible KOR dimer of biological significance.
- It should be made clearer that Nb6 is not specific for KOR, but that it also binds to the other opioid receptors.

Reviewers' comments:

Reviewer #1 (Remarks to the Author):

The authors have very extensively revised the manuscript to address my concerns. I support publication of this manuscript in Nature Communications.

Reviewer #2 (Remarks to the Author):

General comments.

Overall, this is a very impressive work. Getting a crystal structure of any GPCR is difficult, especially in a complex with a Nb. These new tools would be undoubtedly extremely useful for research on KOR and related receptors. The manuscript is well written and easy to follow.

I have reviewed this work before, and the authors have addressed most of my comments in the revised version.

My general comment is that the newly revised parts would benefit from a proofreading. Thanks for the kind advice; all of the newly added parts have been revised and proofed by all authors

My second comment is that some experiments are still not described in sufficient details. This assumes a very high level of expertise from a person who may want to repeat these experiments and telepathic abilities to connect to the author's brain. The gold standard that would really increase the reproducibility of the work would be to write everything in sufficient details so that the first year PhD student could repeat them. Same goes for the details of the data analysis. Simply referring to the software build-in functions may not be sufficient. Not everybody has access to this specific software, nor the software used may have all functions defined.

The Methods section has been modified to include what we hope are sufficiently detailed descriptions that they can be easily replicated. Additionally, all equations used in this manuscript have been included under each assay protocol.

Addressing the following specific comments could further improve the manuscript:

p7. Another available antagonist radioligand 3H-JDTic was also tested, and no significant changes have been found in its binding affinity (K_i) or equilibrium constant (K_d) with KOR in the presence of Nb6 or Nb39 (Fig. 1e, Supplemental Fig. 1a), which makes it an ideal choice to test other KOR ligands.

Fig S1a Shows Kinetic measurements that need to be repeated at several concentrations of the ligand. A standard plot k_{obs} vs ligand concentration should be shown. k_{off} should also be calculated, as well as measured in a separate competition

experiment. This is not dissimilar to the data presented on Fig S7 (although there the results of the competition experiment is shown). A good example could be found here: DOI: 10.1007/978-1-4939-8630-9_10.

We have now repeated the kinetic measurements using three concentrations of radioligand ^3H -JDTic as shown in Figure 1 below. The standard plot 'Kobs v.s. Ligand concentration' has also been shown aside individually. Kon, Koff and Kd values have been summarized in the table right below the plots. This whole figure has also been included as Supplementary Figure S2.

We agree that it would great to use an independent assay (e.g. FRET) to repeat our radioligand binding. However, the lack of fluorescence-labeled KOR ligands prevents us from performing this kind of assay as the reviewer suggested.

Figure 1. An alternative assay to measure JDTic kinetics in the presence of Nb6 or Nb39. The k_{on} was determined by incubation of KOR cell membranes with the indicated concentrations of [3 H]-JDTic for various time periods. Each point is the mean \pm SEM of three experiments. Data were fitted using a one-phase exponential association function to yield an observed on-rate (k_{ob}) (left panel). k_{ob} plotted against the [3 H]-JDTic concentration, employed for indirect determination of k_{off} (y intercept = 0), $K_{on} = (K_{ob} - k_{off})/[ligand]$. (right panel)

Affinity Values and Kinetically Derived Parameters of JDTic

Membranes	k_{on} ($M^{-1} min^{-1}$)	k_{off} (min^{-1})	K_d , nM
KOR + Buffer	1.39×10^8	0.0085	0.061
KOR + Nb6	1.20×10^8	0.0050	0.041
KOR + Nb39	1.22×10^8	0.0117	0.095

The details of the data analysis are missing. A reference to a model in Prizm assumes the reader to have access to the same version of Prizm and that their model is unmodified. This is very detrimental to the reproducibility of the results.

The equations used to plot the graphs have been included in the Methods section.

Reported K_d is rather low to be measured by 3H ligands.

Is this a kinetic K_d ?

The reported values differ by a factor of 10.

The K_d value for JD_{Tic} reported in this manuscript was obtained by measuring the association (K_{on}) and dissociation (K_{off}) rates and then using the standard equation $K_d = K_{off}/K_{on}$. For this determination we used ³H-JD_{Tic} which we believe could produce more reliable kinetic rates for JD_{Tic}. We have also repeated the K_d measurement with an alternative assay as the reviewer suggested (Fig. 1 above), the values are consistent with each other. Moreover, the previously reported K_i of JD_{Tic} (Thomas JB et al., 2001) is 0.3 nM which is consistent with the K_i we obtained in the manuscript (0.2 nM)

In the original paper (Thomas et al, 2001), they mentioned "... demonstrated high affinity for the kappa receptor in the binding assay ($kappa K(i) = 0.3$ nM)"; They also reported **functional** K_i s in [³⁵S]GTP γ S are 0.02 and 0.006 nM which differed from binding affinities in their direct binding assays.

it is also refereed in text as showing no significant different in K_i or K_d but it seems to wrongly referred to as it does not show these data. Did the authors mean S1b?

Here no significant difference in K_i or K_d refers to the comparisons of K_i or K_d values in the presence of buffer, Nb6 or Nb39 for KOR and JD_{Tic}. Based on Figure 1e and Supplementary Figure S1a, there are no significant differences.

p8. N6 co should be Nb6

This error has been corrected.

p10 (Supplemental Fig. 1a). supplementary?

All supplemental has been changed to supplementary.

p13. backbone of L353 in helix α 5 in G α i proteins28. Ideally, CGN numbering (eg, G.H5.25; doi 10.1038/nature14663) should be used for G proteins, or at least the isoform should be indicated as it can be L354 in G α i2 or G α z, for example.

The numbering has been corrected using the CGN numbering methods.

p15. JD_{Tic} also has a slower off-rate in the presence of Nb6 and appears to outcompete the binding of Sal A, occupies the binding pocket and stabilizes the KOR/Nb6 complex.

p15. The relatively slower $t_{1/2}$ for JD_{Tic} in KOR/Nb39 (~107 s) compared to KOR/Nb6

(~75 s) is likely due to the cooperativity from Nb39 because Nb39 has been shown to significantly increase the binding affinity of Sal A (Supplemental Fig. 1b and 7b) S1b does not show any kinetic data. Wrong reference to the figure?

Supplemental Fig. 1b has been removed from this reference.

p15. “relatively fast kinetics”. Relative to what? It is not clear what the authors use as a reference point, or if there any generally accepted values for kinetics that would qualify nanobody’s as conformation-specific tools.

To avoid the confusion, we changed this sentence to “This observed reversible interaction between receptor and nanobody could provide a potential conformation-specific biosensor approach for investigating these transitions for a variety of applications.”

Fig S2a-all panels. Y axis. Looks like the axis title is wrong

Fig S3a Y axis. Wavelength used not shown. A280?

Fig S3b Y axis. Looks like the axis title is wrong

All the missing information has been added to the graphs.

Figure S2. This is unaddressed original comment (ug of what? It has to be clear from the figure or the legend without going through the methods – and this is not explained in the methods well) that the authors do not explain the details of the experiments in sufficient detail. Although they have addressed this in the rebuttal letter they have not incorporated the changes in the manuscript.

Thanks for pointing this out. The “ \$\mu\text{g}\$ ” refers to the mass of nanobody DNA transfected into the cells. This information has been added to the figure legend and Methods.

Reviewer #3 (Remarks to the Author):

The authors previously reported two nanobodies interacting with the kappa opioid receptor (KOR): Nb6 that decreases agonist binding (but interacting with any opioid receptor subtype), and Nb39 that in contrast increases agonist binding and appears specific for KOR. The present paper was aimed at better characterizing these two nanobodies and identify the structural bases for their allosteric effects. Nb6 is shown to bind preferentially an inactive form of KOR, very similar to that stabilized by the antagonist JD_{Tic}, then decreasing agonist potency at displacing 3H-JD_{Tic} binding. In contrast, Nb39 increases antagonist and decreases agonist affinities. The effect of Nb6 was confirmed by functional studies showing a decreased potency and/or efficacy of several KOR agonists. The crystal structure of KOR-JD_{Tic}-Nb6 complex was solved, and revealed an inactive conformation, slightly more compact than that observed with a neutral antagonist, and in which the ECL2 form a lid at the binding site entrance. This structure is then quite different from that reported by the authors for an active form of the receptor stabilized by Nb39. These nanobodies are then used to

detect KOR states in living cells, with Nb39 interaction being favored upon agonist action, while Nb6 interaction is favored in the basal or antagonist bound forms. They eventually extend their use of Nb6 to detect the activation of other class A receptors based on a chimeric receptor assay.

Minor points

- The authors should add a short comment on the presence of two KORs per asymmetric crystal units, indicating the proteins are upside down, then unlikely to represent possible KOR dimer of biological significance.

We have added that comment “This anti-parallel crystal packing should be attributed to the unique crystallization condition, unlikely to represent possible dimerization under physiological conditions.” In the main text.

- It should be made clearer that Nb6 is not specific for KOR, but that it also binds to the other opioid receptors.

We now have a paragraph discussing the application of Nb6 to other opioid receptors (DOR, MOR, and NOP)

“Finally, we note that due to the highly conserved sequence and structure of opioid receptors (Supplementary Fig. 9a), Nb6 also binds to μ , δ and nociceptin opioid receptors with high affinity (Supplementary Fig. 9b). Nb6 thus can also report the activation and inactivation of the nociceptin receptor (Supplementary Fig. 9c), while Nb39 has been shown unable to bind to this receptor¹⁴. Thus, together with Nb39, this nanobody pair provides a toolset for studying the dynamics of conformational transitions *in situ* for other members of the opioid receptor family.”

Reviewers' Comments:

Reviewer #2:

Remarks to the Author:

This is a much improved version of the manuscript and the authors have addressed my comments.